# Fatty acid synthase inhibition alleviates lung fibrosis via β-catenin signal in fibroblasts

Hui Lian[1], Yujie Zhang[1], Zhao Zhu[1], Ruyan Wan[1], Zhixia Wang[2], Kun Yang[1], Shuaichen Ma[1], Yaxuan Wang[1], Kai Xu[1] ⓘ, Lianhui Cheng[1], Wenyu Zhao[1], Yajun Li[1], Lan Wang[1] ⓘ, Guoying Yu[1] ⓘ

**Idiopathic pulmonary fibrosis is a progressive and lethal inter-stitial lung disease with an unclear etiology and limited treatment options. Fatty acid synthase (FASN) plays various roles in metabolic-related diseases. This study demonstrates that FASN expression is increased in fibroblasts from the lung tissues of patients with idiopathic pulmonary fibrosis and in bleomycin-treated mice. In MRC-5 cells, the inhibition of FASN using shRNA or the pharmacological inhibitor C75 resulted in the increased mRNA and protein expression of glycogen synthase kinase 3β and Axin1, both negative regulators of the Wnt/β-catenin signaling pathway, and promoted autophagy. This outcome led to a decrease in β-catenin protein and mRNA levels, effectively inhibiting the proliferation, migration, and differentiation of lung fibroblasts into myofibroblasts, while inducing the differentiation of fibro-blasts into adipofibroblasts. In vivo experiments showed that C75 alleviated bleomycin-induced lung fibrosis in mice by inhibiting β-catenin. In conclusion, these findings suggest that inhibiting FASN in fibroblasts may diminish the activity of the Wnt/β-cat-enin signaling pathway, providing a potential therapeutic avenue for pulmonary fibrosis.**

## Introduction

Idiopathic pulmonary fibrosis (IPF) is a progressive fibrotic lung disease characterized by the impairment of lung physiological function and the aggravation of respiratory symptoms (Podolanczuk et al, 2023). The epidemiological features of IPF suggest that multiple risk factors including age, genetic alterations, and environ-mental factors such as cigarette smoke exposure increase disease susceptibility (Mebratu et al, 2023). Over the last two decades, the median survival for IPF patients has been reported to be 3–5 yr without effective lung transplantation, with survival time decreasing with increasing age and in males (Luppi et al,

2021). Currently, two anti-fibrosis drugs, pirfenidone and nin-tedanib, are approved for the treatment of IPF (Kreuter et al, 2019). However, 20–30% of patients are not tolerated in clinical trials and applications in the long term, mainly because of the gastrointestinal side effects (Gulati & Luckhardt, 2020). This suggests an urgent need for extensive research into the path-ogenesis of IPF, with trials of more effective and well-tolerated agents.

Fatty acid synthase (FASN) is a widely expressed cytosolic en-zyme that initiates de novo lipogenesis, a process involving the synthesis of fatty acids from basic building blocks (Zhang et al, 2021; Zhang et al, 2023a). In adult tissues, FASN is typically expressed at low to moderate levels in normal cells. Previous studies have examined FASN using RNA-sequencing and Western blotting techniques. These studies found that compared with normal controls, both the mRNA and protein expression levels of FASN were reduced in lung tissues of IPF patients, particularly in alveolar epithelial cells, ciliated cells, and macrophages (Qian et al, 2022; Shin et al, 2023; Hayek et al, 2024). However, the restructuring of lung tissue architecture in IPF may lead to varied alterations in FASN expression among different cell types within the lung tissues. During the progression of IPF, fibroblasts serve as the primary effector cells (Staab-Weijnitz, 2022). It is widely recognized that the aberrant activation of multiple signaling pathways in pulmonary fibroblasts, including transforming growth factor-β1 (TGF-β1), hedgehog, and Wnt/β-catenin pathways, contributes to the acti-vation of these cells in IPF, promoting cell proliferation, migration, and the secretion of additional cytokines and matrix proteins (Mackinnon et al, 2012; Chen et al, 2018; Lv et al, 2020). Previous studies have shown that the metabolic demands of normal and pathological cells during processes such as growth and migration are similar (Keibler et al, 2016; Martin-Perez et al, 2022). For instance, during adipogenesis, FASN is protected from degradation in a nutrient-dependent manner through its interaction with O-linked β-N-acetylglucosamine transferase (Raab et al, 2021). In pediatric brain tumor medulloblastoma, an overactive sonic hedgehog signaling pathway promotes tumor progression through E2F

[1]State Key Laboratory of Cell Differentiation and Regulation, Henan International Joint Laboratory of Pulmonary Fibrosis, Henan Center for Outstanding Overseas Scientists of Organ Fibrosis, Pingyuan Laboratory, College of Life Science, Henan Normal University, Xinxiang, China    [2]Department of Pulmonary and Critical Care Medicine, The First Affiliated Hospital of Xinxiang Medical University, Weihui, China

Correspondence: wanglan@htu.edu.cn; guoyingyu@htu.edu.cn

transcription factor 1 and FASN-dependent mechanisms (Bhatia et al, 2012). We hypothesize that FASN expression in activated fibroblasts within the IPF lung tissue may be up-regulated to enhance lipid synthesis, thus providing a survival advantage to these cells. Conversely, inhibiting FASN in fibroblasts may induce a quiescent state, potentially alleviating pulmonary fibrosis.

To test this hypothesis, we investigated FASN expression in fibroblasts derived from the fibrotic lung tissue and examined the specific effects of FASN inhibition on these cells in vitro, exploring the underlying mechanisms. Finally, we applied the pharmacological inhibitor C75 in a bleomycin-induced pulmonary fibrosis mouse model to evaluate its impact on lung fibrosis.

# Results

### The expression of FASN was significantly increased in fibroblasts of IPF and bleomycin-induced mouse lungs

Immunofluorescence staining was initially conducted on lung tissue sections from IPF patients. The results revealed co-localization of FASN and α-smooth muscle actin (α-SMA)–positive fibroblasts in the fibrotic lesions of IPF, whereas FASN was predominantly expressed in type II alveolar epithelial cells in the donor lungs, and no significant co-localization of α-SMA and FASN was observed (Fig 1A). Concurrently, in bleomycin-induced mouse lungs, immunofluorescence staining also indicated co-localization of FASN and α-SMA–positive fibroblasts in the fibrotic lung (Fig 1B). To further validate this finding, MRC-5 cells were stimulated with the well-known profibrotic growth factor TGF-β1, demonstrating a significant increase in the protein and mRNA levels of FASN (Fig 1C–E). In addition, compared with control mouse fibroblasts, the protein expression and mRNA levels of FASN in fibroblasts isolated from bleomycin-induced mouse lungs were up-regulated as well (Fig 1F–H). In summary, FASN expression is significantly elevated in fibroblasts from IPF patients and in bleomycin-induced fibrotic mouse lungs.

### Inhibition of FASN induces fibroblast quiescence

When FASN protein expression was knocked down using FASN shRNA in MRC-5 cells, a significant decrease in cell migration and invasion capabilities was observed (Fig 2A and B). Immunofluorescence staining in MRC-5 cells showed that FASN shRNA significantly increased the fluorescence intensity of perilipin-2 (PLIN2), which was the adipofibroblast marker, and reduced the fluorescence intensity of collagen 1 (Fig 2C and D). The Western blotting and quantitative RT–PCR analysis results demonstrated that FASN shRNA significantly decreased the protein and mRNA expression levels of classic markers associated with fibroblast-to-myofibroblast differentiation including fibronectin, collagen 1, and α-SMA, and increased the protein and mRNA expression levels of PLIN2 (Fig 2E–G). Similarly, inhibiting FASN enzymatic activity with C75 in MRC-5 cells markedly suppressed cell migration and invasion capabilities as well (Fig 2H and I). After C75 treatment of MRC-5 cells, immunofluorescence staining showed consistent results for PLIN2 and collagen 1 (Fig 2J and K). Treatment of MRC-5 cells with C75

yielded comparable results in Western blotting and quantitative RT–PCR analysis to those observed with FASN shRNA (Fig 2L–N). In conclusion, FASN inhibition kept fibroblasts in a quiescent state and induced fibroblast lipogenic differentiation.

### β-Catenin promotes the FASN expression

The results of immunofluorescence staining in MRC-5 cells revealed that the overexpression of β-catenin led to increased fluorescence intensity of FASN, whereas the vector control did not produce such an effect (Fig 3A). Furthermore, the Western blotting and quantitative RT–PCR analyses demonstrated that in MRC-5 cells, the overexpression of β-catenin significantly increased both the protein and mRNA expression levels of FASN (Fig 3B–D). Similar findings were observed in HEK293T cells (Fig 3E–G). Furthermore, immunoprecipitation revealed an interaction between β-catenin, glycogen synthase kinase 3B (GSK3B), Axin1, and FASN in HEK293T cells, whereas casein kinase 1a (CK1a) did not exhibit interaction (Fig 3H). Therefore, β-catenin can up-regulate FASN expression through its interaction with FASN.

### FASN inhibition significantly reduced the expression of β-catenin

Immunofluorescence staining results demonstrated that FASN shRNA decreased the fluorescence intensity of β-catenin in MRC-5 cells. Upon the addition of the β-catenin activator, laduviglusib (Ladu), there was a notable increase in the fluorescence intensity of β-catenin in both the cytoplasm and the nucleus, whereas FASN shRNA further attenuated the fluorescence intensity of β-catenin in these cellular compartments (Fig 4A). Subsequent Western blotting analysis in MRC-5 cells demonstrated FASN shRNA significantly reduced the total protein level of β-catenin (Fig 4B and C). Similar findings were observed in fibroblasts isolated from mouse lungs (Fig S1A–C). In addition, Western blotting results indicated that FASN shRNA notably decreased the content of β-catenin in both the cytoplasm and the nucleus of MRC-5 cells (Fig 4D–F). Treatment of MRC-5 cells with C75 and Ladu yielded immunofluorescence staining results for β-catenin consistent with those observed with shFASN (Fig 4G). Furthermore, Western blotting analysis demonstrated a significant reduction in the protein content of β-catenin in MRC-5 cells after treatment with C75 (Fig 4H and I). Similar outcomes were observed in fibroblasts isolated from mouse lungs (Fig S1D–F). Additional nuclear–cytoplasmic fractionation and Western blotting results confirmed that C75 significantly reduced the content of β-catenin in both the cytoplasm and the nucleus of MRC-5 cells (Fig 4J–L). Quantitative RT–PCR analysis results demonstrated that inhibition of FASN by shRNA decreased the mRNA expression levels of FASN and β-catenin in MRC-5 cells (Fig 4M), and C75 decreased the mRNA expression levels of β-catenin in MRC-5 cells (Fig 4N). Overall, both knockdown of FASN expression and inhibition of its enzymatic activity significantly reduced β-catenin levels at the transcriptional and protein stages.

### Inhibition of FASN increased the expression of GSK3B and Axin1

In MRC-5 cells, inhibition of FASN with shRNA and C75 resulted in increased mRNA levels of GSK3B and Axin1 (Fig 5A and B), as well as

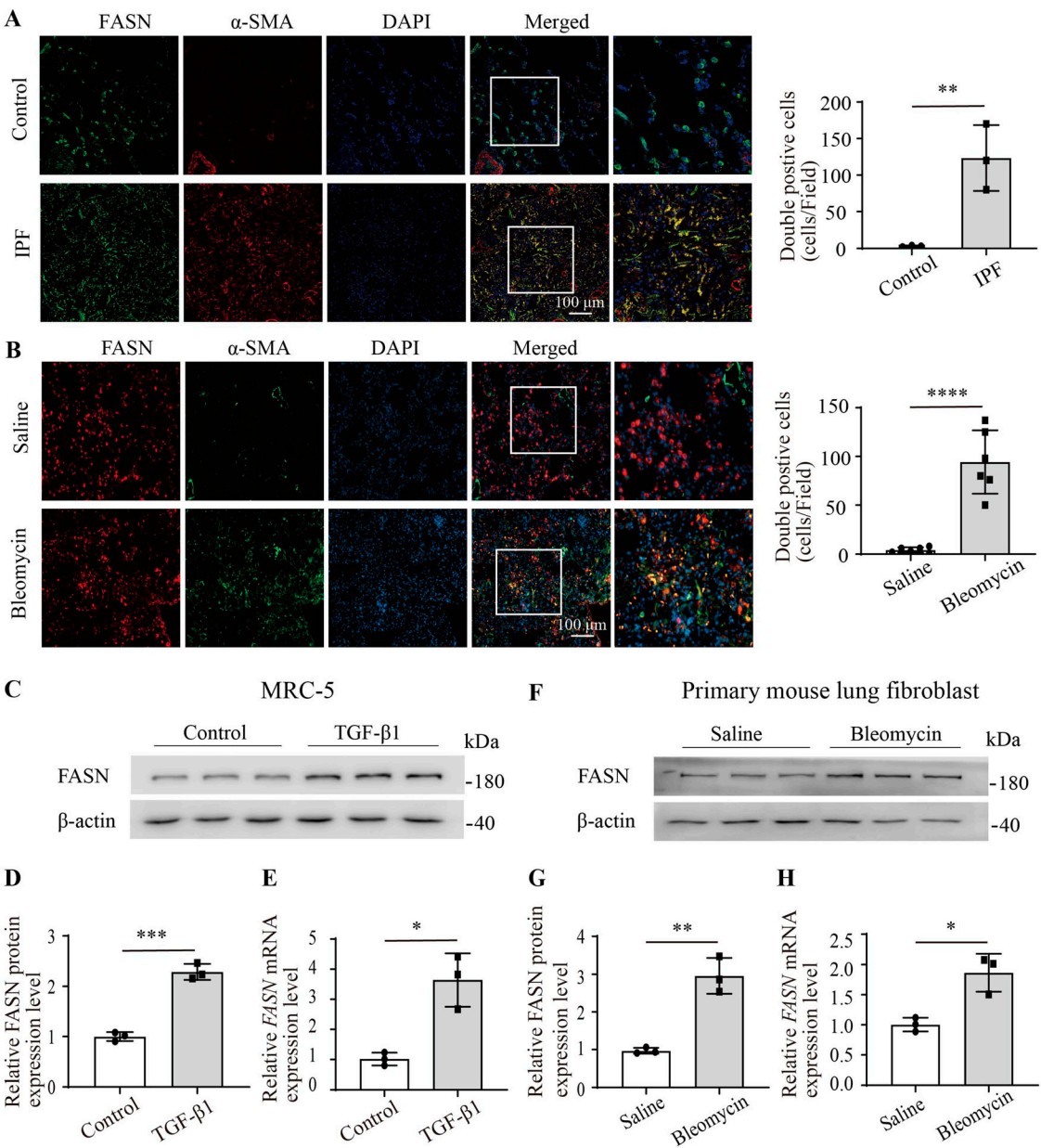

**Figure 1. Increased expression of FASN in lung fibroblasts of IPF and bleomycin-induced fibrotic mouse.**
**(A)** Representative images and statistical analysis of double-labeled immunofluorescence staining for FASN and α-SMA in lung sections from IPF patients (n = 3) and donors (n = 3). **(B)** Representative images and statistical analysis of double-labeled immunofluorescence staining for FASN and α-SMA in lung sections from bleomycin-induced fibrotic mouse (n = 6) and saline controls (n = 6). Figures inside the white box were zoomed in on the rightmost. **(C, D)** Western blotting analysis of FASN protein expression in MRC-5 cells stimulated with or without TGF-β1 (10 ng/ml) for 24 h. n = 3. **(E)** Quantitative RT–PCR analysis of relative *FASN* mRNA expression of MRC-5 cells stimulated with or without TGF-β1 (10 ng/ml) for 12 h. n = 3. **(F, G)** Western blotting analysis of FASN protein expression in primary lung fibroblasts isolated from bleomycin-induced fibrotic mouse and saline mouse. n = 3. **(H)** Quantitative RT–PCR analysis of the relative *FASN* mRNA expression of primary lung fibroblasts isolated from bleomycin-induced fibrotic mouse and saline mouse. n = 3.
Source data are available for this figure.

elevated relative protein expression levels of GSK3B (Fig 5C and D) and Axin1 (Fig 5E and F). In HEK293T cells, inhibition of FASN with shRNA and C75 also resulted in increased mRNA levels of *GSK3B* and *Axin1* (Fig 5G and H), as well as elevated relative protein expression levels of GSK3B (Fig 5I and J) and Axin1 (Fig 5K and L). Therefore, inhibiting FASN led to the increased expression of GSK3B and Axin1.

### Inhibiting FASN suppressed proliferation and enhanced the degradation of β-catenin by increasing autophagic flux

EdU and Cell Counting Kit-8 (CCK8) assays revealed that FASN shRNA suppressed cell proliferation and viability in MRC-5 cells (Fig 6A–C). Treatment with cycloheximide (CHX) significantly promoted

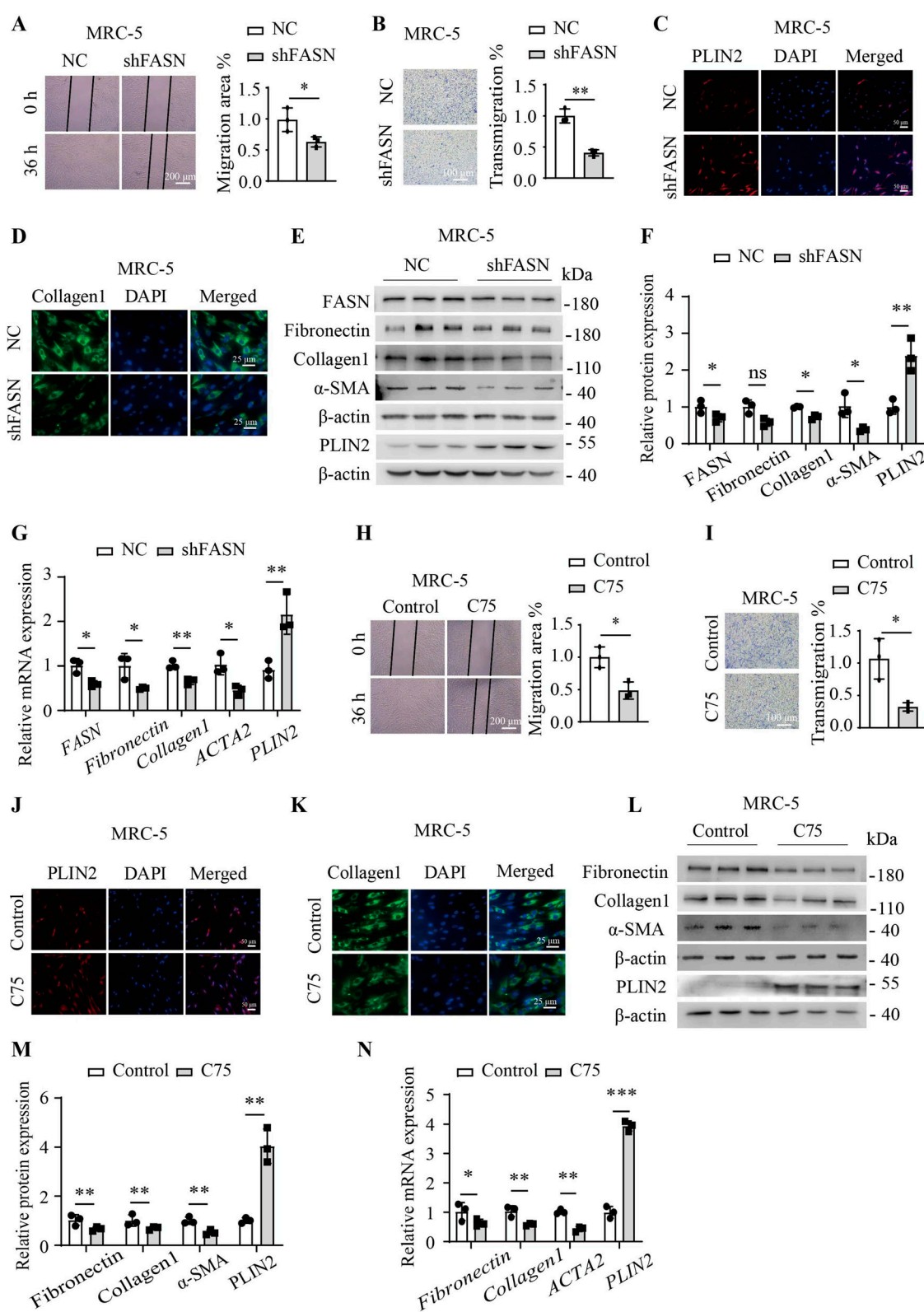

**Figure 2. Inhibition of FASN-induced fibroblasts in a quiescent state.**
**(A)** Images show the migration area of MRC-5 cells by the wound-healing assay after FASN shRNA (n = 3) or NC (n = 3). **(B)** Images show transmigration cells down to the Transwell chamber with FASN shRNA or NC. **(C)** Representative image of immunofluorescence staining for PLIN2 in MRC-5 cells by FASN shRNA (n = 3) or NC (n = 3). **(D)** Representative image of immunofluorescence staining for collagen 1 in MRC-5 cells by FASN shRNA (n = 3) or NC (n = 3). **(E, F)** Western blotting analysis of the protein expression of FASN, fibronectin, collagen 1, α-SMA, and PLIN2 in MRC-5 cells by FASN shRNA (n = 3) or NC (n = 3). **(G)** Quantitative RT–PCR analysis of relative *FASN*,

β-catenin degradation after treatment with FASN shRNA (Fig 6D). Subsequently, Western blotting analysis revealed a significant increase in LC3 II/LC3 I in the FASN shRNA group compared with the NC group after chloroquine (CQ) treatment (Figs 6E and S2A). Further co-transfection of the GFP-RFP-LC3 plasmid with FASN shRNA or NC into MRC-5 cells followed by CQ treatment showed a significant decrease in GFP/RFP fluorescence in the FASN shRNA group compared with the NC group (Figs 6F and S2C), indicating that FASN shRNA markedly promoted autophagy. In MRC-5 cells, EdU and CCK8 assay results demonstrated that C75 inhibited cell proliferation and viability (Fig 6G–I). Treatment of MRC-5 cells with CHX followed by C75 treatment also significantly promoted the degradation of β-catenin (Fig 6J). In addition, pre-treatment or no pre-treatment of MRC-5 cells with C75 followed by CQ treatment resulted in a significant increase in LC3 II/LC3 I compared with the control group (Figs 6K and S2B). After transfection of MRC-5 cells with the GFP-RFP-LC3 plasmid for 24 h, the C75 and CQ treatment groups showed a decrease in GFP/RFP fluorescence compared with the control group (Figs 6L and S2D), indicating that C75 also promoted autophagy. In summary, these experimental findings indicate that FASN inhibition enhances autophagic flux and promotes the degradation of β-catenin, while concurrently suppressing cell proliferation.

### C75 alleviated pulmonary fibrosis in bleomycin-induced mouse lungs

To assess the function of FASN on the process of fibrogenesis in vivo, a mouse model of pulmonary fibrosis was established using intratracheal administration of bleomycin. Subsequent to a solitary administration of bleomycin on day 0, mice were subjected to daily intraperitoneal injections of the FASN pharmacological inhibitor C75 or saline for a duration of 10 d. On day 21, mouse lungs were collected (Fig 7A). The survival analysis of mice showed that C75 increased the survival rate of bleomycin-induced fibrotic mice (Fig 7B). Micro-CT scanning revealed that C75 ameliorated the structural changes in bleomycin-induced pulmonary fibrosis; hematoxylin and eosin staining, and Masson's trichrome staining demonstrated that C75 significantly reduced the fibrotic area in bleomycin-induced pulmonary fibrosis (Fig 7C). The acute lung injury score revealed a significant increase in bleomycin-induced pulmonary fibrosis mice, whereas C75 significantly decreased the score in bleomycin-induced pulmonary fibrosis mice (Fig 7D). The hydroxy-proline assay found that C75 decreased the hydroxyproline content in bleomycin-induced mouse lungs (Fig 7E). Immunohistochemical staining for collagen 1 indicated that C75 significantly reduced collagen 1 deposition in bleomycin-induced mouse lungs (Fig 7F and G). Western blotting analysis further confirmed that C75 significantly reduced the protein levels of fibronectin, collagen 1,

α-SMA, p-β-catenin (Ser675), and β-catenin in the lung tissues of bleomycin mice, whereas it simultaneously increased the protein expression levels of Plin2 (Fig 7H–N). Immunofluorescence staining demonstrated a reduction in the fluorescence intensity of α-SMA in the lung tissues of bleomycin-treated mice after C75 treatment, alongside an increase in the fluorescence intensity of Plin2 (Fig 7O). At the same time, immunofluorescence staining also demonstrated a marked reduction in the fluorescence intensity of p-β-catenin (Ser675) and β-catenin in the lung tissues of bleomycin-treated mice after C75 treatment (Fig 7P and Q). In addition, the Lipi-Deep Red probe was used to detect the lipid droplet content in mouse lung tissues. Compared with the saline group, the fibrotic regions in the bleomycin group showed a significant reduction in the lipid droplet content, whereas lipid droplet levels were relatively increased in bleomycin-treated lung tissues after C75 treatment (Fig 7R). In conclusion, C75 alleviated bleomycin-induced pulmonary fibrosis.

## Discussion

In the present study, immunofluorescence staining demonstrated the co-localization of both FASN and α-SMA–positive fibroblasts in the fibrotic lesions of IPF and bleomycin-induced fibrotic mice. Moreover, stimulation with TGF-β1 significantly elevated the expression of FASN in MRC-5 cells. Furthermore, compared with control mouse fibroblasts, FASN expression was up-regulated in fibroblasts isolated from bleomycin-induced fibrotic lungs. This observation suggested FASN expression was significantly elevated in fibroblasts from IPF patients and bleomycin-induced fibrotic mouse lungs.

It is well documented that abnormal activation of fibroblasts leads to increased cell migration and invasion, as well as excessive ECM deposition, which play significant roles in the pathogenesis and progression of pulmonary fibrosis (Yu et al, 2018). Adipofibroblasts, situated adjacent to alveolar epithelial cells, represent a quiescent population of mesenchymal fibroblasts that have been thoroughly characterized in neonatal rodents and confirmed through lineage tracing analyses after alveolar injury (Li et al, 2018; Trempus et al, 2023). The fibrotic response induced by bleomycin is reversible, allowing researchers to explore the fate of activated fibroblasts during fibrosis resolution. Results indicated that during this resolution, activated fibroblasts and their progeny were not cleared from the lungs, with some cells dedifferentiating into an adipofibroblast phenotype (Zhang et al, 2023b). Adipofibroblasts in the alveoli that express signal peptide–CUB–epidermal growth factor–like domain–containing protein 2 also express the lipid droplet marker PLIN2 (Tsukui et al, 2020, 2024). In the current study, both FASN shRNA and C75 effectively inhibited

---

*fibronectin, collagen 1, ACTA2,* and *PLIN2* mRNA expression in MRC-5 cells with FASN shRNA (n = 3) or NC (n = 3). **(H)** Images show the migration area of MRC-5 cells by the wound-healing assay by C75 (30 μM, n = 3) for 36 h or not (n = 3). **(I)** Images show transmigration cells down to the Transwell chamber with C75 (30 μM, n = 3) for 48 h or not (n = 3). **(J)** Representative images of immunofluorescence staining for PLIN2 in MRC-5 cells by C75 (30 μM, n = 3) for 48 h or not (n = 3). **(K)** Representative images of immunofluorescence staining for collagen 1 in MRC-5 cells by C75 (30 μM, n = 3) for 48 h or not (n = 3). **(L, M)** Western blotting analysis of the protein expression of fibronectin, collagen 1, α-SMA, and PLIN2 in MRC-5 cells by C75 (30 μM, n = 3) for 48 h or not (n = 3). **(N)** Quantitative RT–PCR analysis of relative *fibronectin, collagen 1, ACTA2,* and *PLIN2* mRNA expression in MRC-5 cells without or with C75 (30 μM, n = 3) for 12 h or not (n = 3).
Source data are available for this figure.

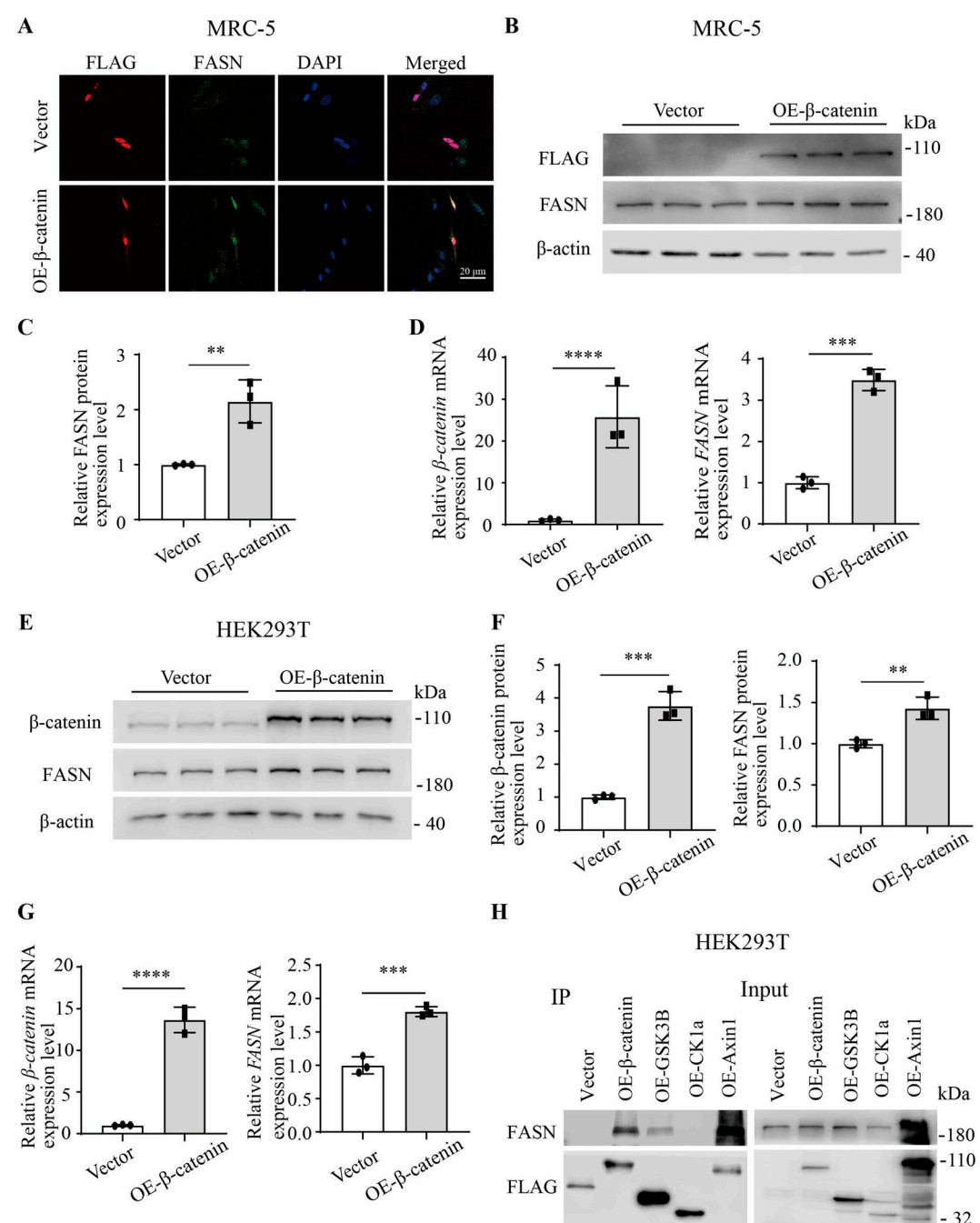

**Figure 3. β-Catenin promoted the expression of FASN.**
**(A)** Representative images of immunofluorescence staining illustrating the co-localization of overexpression (OE)-β-catenin (n = 3) or not OE-β-catenin (n = 3) and FASN in MRC-5 cells. **(B, C)** Western blotting analysis of FLAG and FASN expression in MRC-5 cells by OE-β-catenin by Phage-FLAG-β-catenin or not. **(D)** Quantitative RT–PCR analysis of relative *β-catenin* and *FASN* mRNA expression in MRC-5 cells by OE-β-catenin (n = 3) or not (n = 3). **(E, F)** Western blotting analysis of β-catenin and FASN expression in HEK293T cells by OE-β-catenin (n = 3) or not (n = 3). **(G)** Quantitative RT–PCR analysis of relative *β-catenin* and *FASN* mRNA expression in HEK293T cells by OE-β-catenin (n = 3) or not (n = 3). **(H)** Immunoprecipitation showed that FASN interacted with β-catenin, GSK3B, and Axin1 in HEK293T cells, but not with CK1a. Source data are available for this figure.

the migration and invasion capabilities of fibroblasts, significantly reducing the protein and mRNA expression levels of fibronectin, collagen 1, and α-SMA, while increasing PLIN2 expression. This suggests that FASN inhibition maintains fibroblasts in a quiescent state and induces lipogenic differentiation. In addition, in vivo

experiments demonstrated that C75 alleviated bleomycin-induced pulmonary fibrosis.

Previous investigations have revealed a significant increase in the expression of β-catenin in IPF fibroblasts (Chilosi et al, 2003; Shi et al, 2017). Essentially, fibroblast activation triggered by the Wnt/

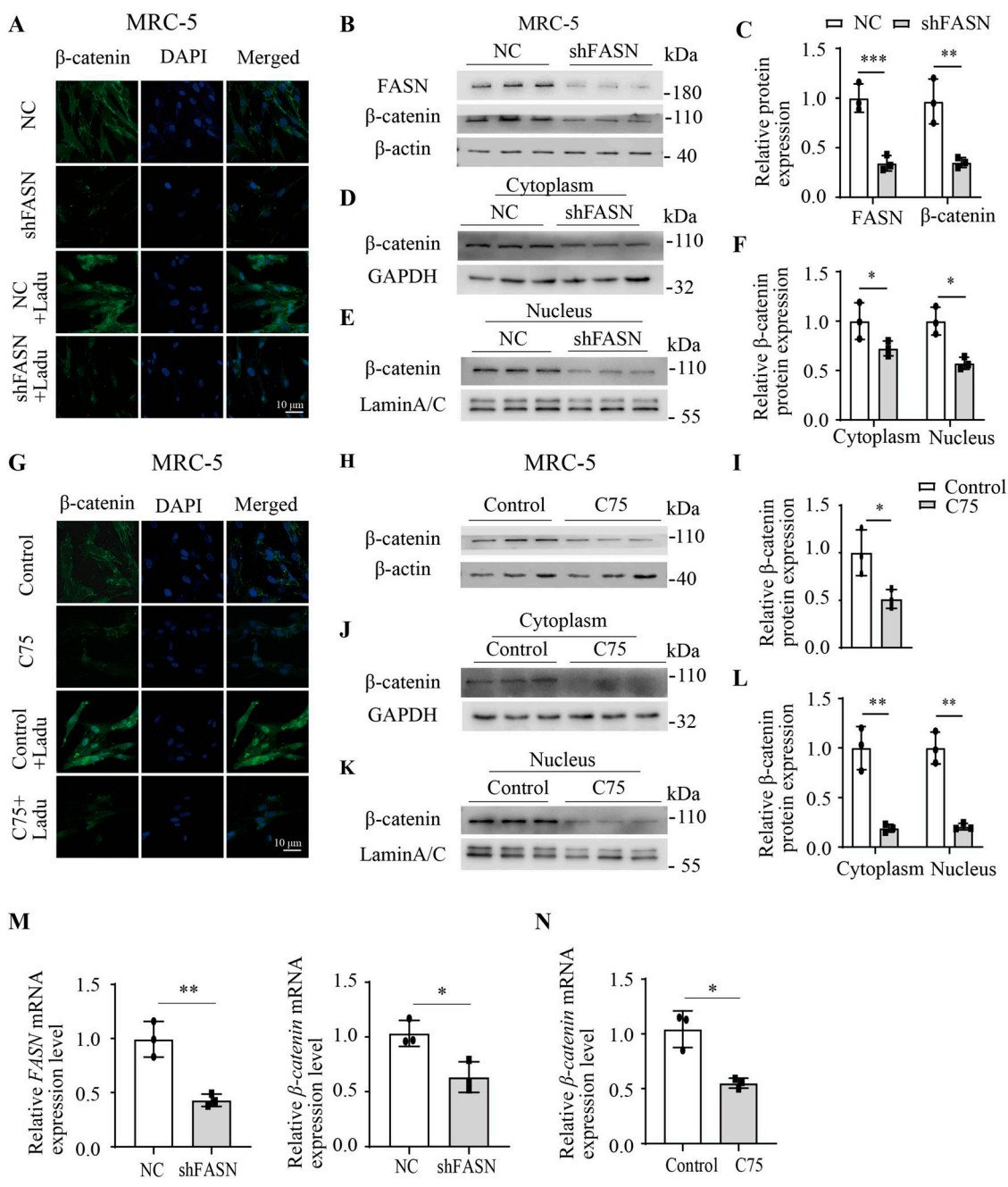

**Figure 4. Inhibition of FASN reduced the expression of β-catenin in fibroblasts.**
**(A)** Representative images of immunofluorescence staining for β-catenin in MRC-5 cells by FASN shRNA or NC and treated with Ladu (5 ng/ml) for 24 h. n = 3.
**(B, C)** Western blotting analysis of the protein expression of FASN and β-catenin in MRC-5 cells by FASN shRNA (n = 3) or NC (n = 3). **(D, E, F)** Western blotting analysis of β-catenin expression in the cytoplasm and in the nucleus in MRC-5 cells by FASN shRNA (n = 3) or NC (n = 3). **(G)** Representative images of immunofluorescence staining for β-catenin in MRC-5 cells by C75 (30 μM) for 24 h and then treated with Ladu (5 ng/ml) for 24 h. n = 3. **(H, I)** Western blotting analysis of the protein expression of β-catenin in MRC-5 cells by C75 (30 μM, n = 3) for 48 or not (n = 3). **(J, K, L)** Western blotting analysis of β-catenin expression in the cytoplasm and in the nucleus in MRC-5 cells with or without C75 (30 μM) for 48 h. n = 3. **(M)** Quantitative RT–PCR analysis of the relative mRNA expression of FASN and β-catenin in MRC-5 cells by FASN shRNA (n = 3) or NC (n = 3). **(N)** Quantitative RT–PCR analysis of the relative mRNA expression of β-catenin in MRC-5 cells by C75 (30 μM, n = 3) for 12 h or not (n = 3).
Source data are available for this figure.

β-catenin signaling pathway requires increased nutrient uptake and energy consumption. Previous studies have highlighted the crucial roles of cellular nutrient and energy requirements in promoting the expression of FASN (Menendez, 2010; Hosseini et al, 2015). FASN is known to facilitate nucleic acid, protein, and lipid synthesis to support cancer cell metabolism and is implicated in various cellular functions such as immune evasion and resistance to cell death (He et al, 2020). In certain cancer cells, FASN has been

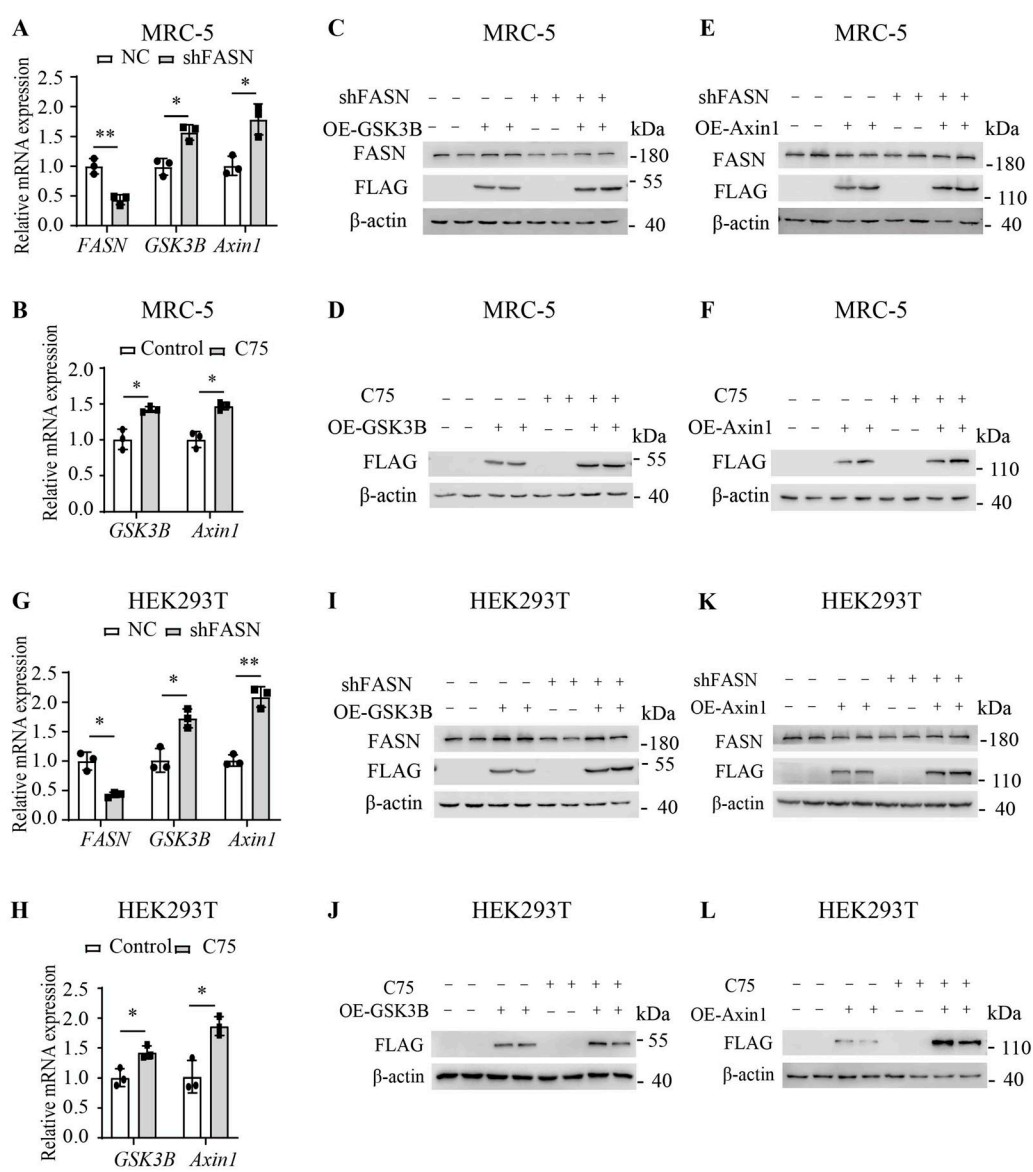

**Figure 5. Inhibition of FASN increased the relative expression of GSK3B and Axin1.**
**(A)** Quantitative RT–PCR analysis of the relative mRNA expression levels of *FASN*, *GSK3B*, and *Axin1* in MRC-5 cells transduced with FASN shRNA (n = 3) or NC (n = 3). **(B)** Quantitative RT–PCR analysis of the relative mRNA expression levels of *GSK3B* and *Axin1* in MRC-5 cells after treatment with C75 (30 μM) for 12 h. n = 3. **(C)** Western blot analysis of GSK3B and FASN protein expression in MRC-5 cells from the FASN shRNA (n = 3) or NC groups (n = 3). **(D)** Western blot analysis of the relative protein expression level of GSK3B in MRC-5 cells after treatment with C75 (30 μM) for 24 h. n = 3. **(E)** Western blot analysis of Axin1 and FASN protein expression in MRC-5 cells from the FASN shRNA (n = 3) or NC groups (n = 3). **(F)** Western blot analysis of the relative protein expression level of Axin1 in MRC-5 cells after treatment with C75 (30 μM) for 24 h. n = 3. **(G)** Quantitative RT–PCR analysis of the relative mRNA expression levels of *FASN*, *GSK3B*, and *Axin1* in HEK293T cells transduced with FASN shRNA (n = 3) or NC (n = 3). **(H)** Quantitative RT–PCR analysis of the relative mRNA expression levels of *GSK3B* and *Axin1* in HEK293T cells after treatment with C75 (30 μM) for 12 h. n = 3. **(I)** Western blot analysis of GSK3B and FASN protein expression in HEK293T cells from the FASN shRNA (n = 3) or NC groups (n = 3). **(J)** Western blot analysis of the relative protein expression level of GSK3B in HEK293T cells after treatment with C75 (30 μM) for 24 h. n = 3. **(K)** Western blot analysis of Axin1 and FASN protein expression in HEK293T cells from the FASN shRNA (n = 3) or NC groups (n = 3). **(L)** Western blot analysis of the relative protein expression level of Axin1 in HEK293T cells after treatment with C75 (30 μM) for 24 h. n = 3. GSK3B or Axin1 was overexpressed using phage-FLAG-GSK3B or phage-FLAG-Axin1. Flag-tagged proteins indicate the corresponding protein levels. Source data are available for this figure.

shown to indirectly regulate the Wnt/β-catenin signaling pathway. For instance, fatty acid–binding protein 5 directly interacted with FASN, thereby regulating its expression and promoting lipid droplet deposition and activation of the Wnt/β-catenin signaling pathway (Lu et al, 2023). In salivary adenoid cystic carcinoma cells, FASN inhibition attenuated invasion, metastasis, and epithelial–mesenchymal transition in a PRRX1/Wnt/β-catenin dependent manner (Zhang et al, 2020). However, there is currently no evidence demonstrating whether β-catenin can regulate FASN expression in IPF fibroblasts or other cell types. Therefore, we investigated whether β-catenin promoted the expression of FASN in both MRC-5 cells and HEK293T cells. The results revealed that β-catenin significantly

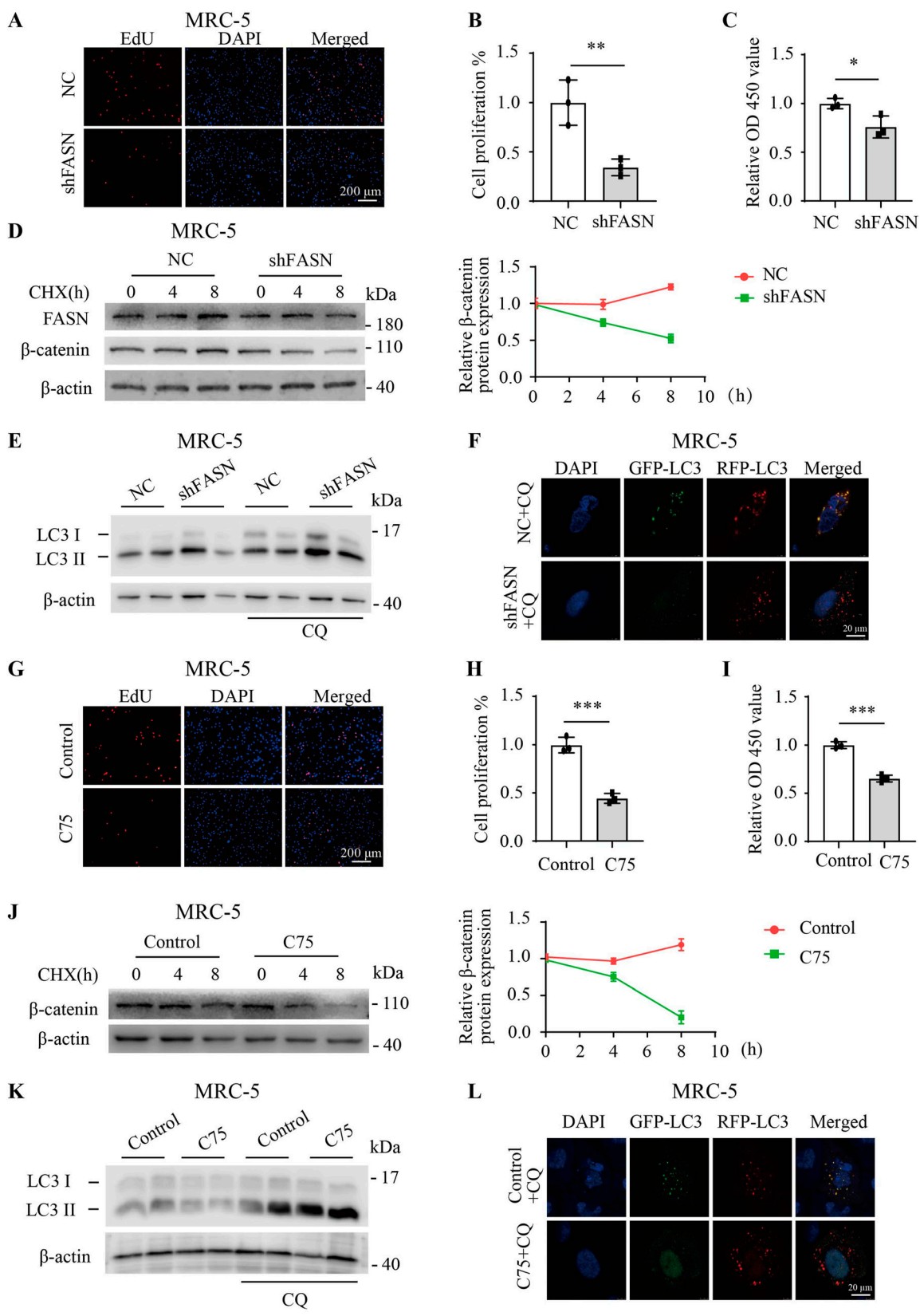

up-regulated the protein and mRNA expression of FASN. This finding suggests that β-catenin also played a crucial role in promoting FASN expression, and this effect was attributed to the direct interaction between β-catenin and FASN.

After various lung injuries in vivo and in vitro, β-catenin, as a central mediator, accumulates in the cytosol and translocates to the nucleus, where it forms a complex with lymphoid enhancer–binding factor 1 to regulate the transcription and translation of ECM-related genes (Song et al, 2018). Given the important role of β-catenin in cytoplasmic aggregation and nuclear translocation in its function, our study also investigated whether FASN shRNA or C75 affected the content of β-catenin in the cytoplasm and nucleus. The results revealed that FASN shRNA or C75 significantly reduced the total protein content and mRNA level of β-catenin in fibroblasts. Furthermore, upon fractionating the cytoplasm and nucleus, this study found that FASN shRNA or C75 decreased the β-catenin content in both compartments. This indicates that FASN inhibition regulates β-catenin expression at both transcriptional and protein levels, respectively.

GSK3B and Axin1 are key negative regulators in the Wnt/β-catenin pathway. Acting as a substrate for GSK3B, Axin1 is phosphorylated by GSK3B, which enhances its binding to β-catenin (Hida et al, 2012). Established experimental evidence has shown that the increased expression of GSK3B and Axin1 notably reduced β-catenin expression in cells (Hida et al, 2012). To further elucidate the precise mechanism by which FASN inhibited the expression of β-catenin, we hypothesized that FASN inhibition may regulate some negative regulators of the Wnt/β-catenin pathway, such as GSK3B or Axin1. In this experiment, both quantitative RT–PCR analysis and Western blotting results showed that after inhibiting FASN with shRNA and C75, the relative mRNA and protein expression levels of GSK3B and Axin1 remained consistently elevated. This may enhance the degradation of β-catenin. Further immunoprecipitation results obtained from HEK293T cells revealed that this effect was due to FASN directly interacted with GSK3B and Axin1.

In addition, previous studies have also demonstrated that β-catenin can interact with LC3 itself and was directly degraded through the autophagy pathway (Wu et al, 2019). It has been shown that changes in membrane lipids can disrupt membrane fluidity, thereby interfering with the normal fusion of organelles such as autophagosomes and lysosomes (Rowland & Czech, 2023). The de novo lipogenesis pathway serves as the primary source of phospholipid fatty acids for autophagosomal membrane synthesis, whereas fatty acids in circulating lipoproteins contribute to lipid storage (Rowland et al, 2023). In adipocytes, the reduction in the fatty acid content resulting from FASN deficiency disrupts both

in vitro and in vivo autophagic flux and lysosomal function (Rowland et al, 2023). However, some experimental results suggest that FASN inhibition promoted autophagy. For instance, FASN inhibitors enhance the accumulation of LC3 II in gastrointestinal stromal tumors (Li et al, 2017). Moreover, although inhibiting FASN expression decreases fatty acid levels in non–small-cell lung cancer, it paradoxically increases the ratio of LC3 II to LC3 I, thereby promoting autophagy (Yan et al, 2021). This suggests that the impact of FASN knockdown on autophagy may vary among different cell types. In this study, both FASN knockdown and inhibition in fibroblasts increased the ratio of LC3 II to LC3 I, promoting autophagic flux and effectively degrading β-catenin protein.

Peroxisome proliferator–activated receptor γ (PPARγ) serves as a master regulator for the lipogenic differentiation of preadipocytes and mesenchymal stem cells and is also involved in the formation of adipofibroblasts (Taniguchi et al, 2008). PPARγ agonists have been shown to protect mice from developing fibrosis (Speca et al, 2021). Research has demonstrated that inhibiting the Wnt signaling pathway results in the increased expression of PPARγ and its target genes, fatty acid–binding protein 2 and PLIN2 (Kim et al, 2019). Our in vitro and in vivo experiments revealed that the inhibition of FASN in fibroblasts can reduce the expression of β-catenin. Therefore, we speculate that this reduction in β-catenin may facilitate the dedifferentiation of myofibroblasts into adipofibroblasts by activating PPARγ, and the specific mechanisms require further investigation.

In summary, our findings indicate that FASN inhibition kept fibroblasts in a quiescent state and induced fibroblast lipogenic differentiation, which effectively alleviated pulmonary fibrosis. Consequently, targeting FASN inhibition in fibroblasts presents a promising therapeutic strategy for pulmonary fibrosis.

# Materials and Methods

### Reagents

Recombinant human TGF-β1 was purchased from R&D Systems. Bleomycin sulfate was purchased from Hisun Pharm. C75, CQ, CHX, and Ladu were purchased from MCE. Lipi-Deep Red was purchased from DOJINDO.

### Isolation of mouse primary lung fibroblasts

Mouse lung fibroblasts from control and bleomycin-treated mice were obtained following the previously established procedures (Wan et al, 2023). Briefly, isolated lungs were minced and digested in

**Figure 6. Inhibition of FASN suppressed proliferation and promoted the degradation of β-catenin by increasing autophagic flux.**
**(A, B)** Representative images of the EdU assay and analysis of MRC-5 cells with FASN shRNA (n = 3) or NC (n = 3). **(C)** Bar graph showed the cell viability measured by CCK8 with FASN shRNA (n = 3) or NC (n = 3). **(D)** Western blotting analysis of β-catenin protein degradation after treatment with 100 μg/ml CHX for different lengths of time in MRC-5 cells than pre-treatment with FASN shRNA (n = 3) or NC (n = 3). **(E)** Western blotting analysis of the protein expression of LC3 II/LC3 I in MRC-5 cells treated with CQ (50 μM) for 4 h after pre-treatment with FASN shRNA (n = 3) or NC (n = 3). **(F)** Immunofluorescence staining of GFP/RFP-LC3 in MRC-5 cells treated with CQ (50 μM) for 4 h after pre-treatment with FASN shRNA (n = 3) or NC (n = 3). **(G, H)** Representative images of the EdU assay and analysis of MRC-5 cells with 30 μM C75 (n = 3) for 24 h or not (n = 3). **(I)** Bar graph showed the cell viability measured by CCK8 with C75 (n = 3) for 24 h or not (n = 3). **(J)** Western blotting analysis of β-catenin protein degradation after treatment with 100 μg/ml CHX for different lengths of time in MRC-5 cells than pre-treatment with C75 (n = 3) for 24 h or not (n = 3). **(K)** Western blotting analysis of the protein expression of LC3 II/LC3 I in MRC-5 cells treated with CQ (50 μM) for 4 h after pre-treatment with C75 (n = 3) for 24 h or not (n = 3). **(L)** Immunofluorescence staining of GFP/RFP-LC3 in MRC-5 cells treated with (CQ, 50 μM) for 4 h after pre-treatment with C75 (n = 3) for 24 h or not (n = 3).
Source data are available for this figure.

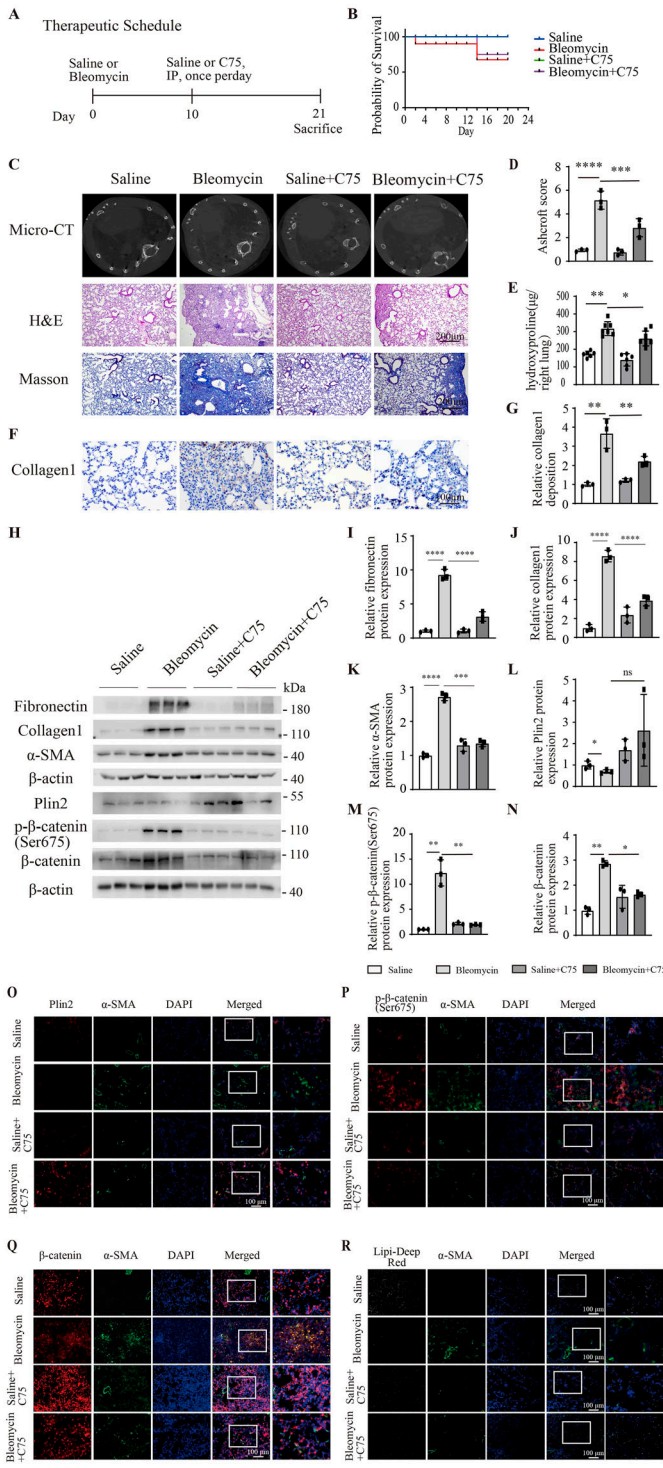

**Figure 7. C75 alleviated pulmonary fibrosis in bleomycin-induced mouse lungs.**
**(A)** Schematic illustrating the experimental procedure for the in vivo administration of bleomycin and C75. **(B)** Survival curve of the bleomycin-induced fibrotic mice treated with C75. n = 8–12. **(C)** Representation images of micro-CT, hematoxylin and eosin staining, and Masson's staining in bleomycin-induced fibrotic mice treated with C75. n = 3. **(D)** Analysis of the Ashcroft score of the bleomycin-induced fibrotic mice treated with C75 according to hematoxylin and eosin staining. n = 3. **(E)** Bar graphs show the right lung hydroxyproline content measured at day 21 in saline and C75-treated mice challenged with bleomycin or not. n = 6–8. **(F, G)** Immunohistochemical staining for collagen 1 and

collagenase buffer at 37°C for 30 min. Subsequently, the cell pellets were centrifuged at 500*g* per minute at room temperature for 10 min. After the removal of the supernatant, the cell pellets were resuspended and cultured for ~6 d. Mouse lung fibroblasts were cultured in complete MEM-Low medium (Procell), supplemented with 10% FBS, 100 IU penicillin, and 100 μg/ml streptomycin.

## Cell culture and treatment

The MRC-5 cell line was procured from the American Type Culture Collection and cultured in complete MEM (Procell) supplemented with 10% FBS under standard culture conditions. HEK293T cells were obtained from Procell and cultured in complete DMEM-High medium (Procell) supplemented with 10% FBS under standard culture conditions.

When MRC-5 cells reached ~90% confluence, they were detached using 0.25% trypsin/EDTA and seeded into six-well plates at 40% confluence. After cell attachment, the adherent cells were serum-starved for 6 h in MEM supplemented with 1% FBS, followed by treatment with various compounds TGFβ1 (10 ng/ml), C75 (30 μM), or Ladu (5 ng/ml) in complete MEM for 12, 24, or 48 h.

## Plasmid construction, lentivirus package, and stable infected cell line generation

The overexpression β-catenin, GSK3B, CK1a, and Axin1 plasmids of Phage-FLAG-β-catenin, Phage-FLAG-GSK3B, Phage-FLAG-CK1a, and Phage-FLAG-Axin1 or the empty Phage-FLAG plasmids were synthesized and transfected into HEK293T cells by Lipofectamine 3000 according to the manufacturer's instructions. Human FASN shRNA (Targeted 5′-CCTACTGGATGCGTTCTTCAA-3′) and mouse FASN shRNA (Targeted 5′-GCTGGTCGTTTCTCCATTAAA-3′) were designed and synthesized by Sangon Biotech and subsequently annealed and inserted into the pLKO.1. For the lentivirus package, HEK293T cells were seeded into a 10-cm dish with 30% confluent; after 12 h, the plasmids of shFASN, NC, Phage-FLAG-β-catenin, and Phage-FLAG (10 μg), psPAX2 (4.5 μg), and MD2G (1.5 μg) were transfected. The lentivirus supernatant was separately collected at 48 and 72 h, then concentrated by a centrifugal filter unit. The concentrated lentivirus particles were used to infect 30% subconfluent MRC-5 cells or the fibroblast isolated from mouse lungs in the presence of 2.5 μg/ml

analysis of the positive area with ImageJ to show collagen deposition in the lung tissue. n = 3. **(H)** Western blotting analysis of the protein expression levels of fibronectin, collagen I, α-SMA, p-β-catenin (Ser675), β-catenin, and Plin2 in saline and C75-treated mice, with or without bleomycin challenge. n = 3. **(H, I, J, K, L, M, N)** Statistical analysis of the grayscale values of fibronectin, collagen I, and α-SMA as shown in panel (H). **(O)** Representative images of immunofluorescence staining for Plin2 and α-SMA in lung sections from bleomycin-induced fibrotic mice, treated with or without C75. n = 3. **(P)** Representative images of immunofluorescence staining attaining for p-β-catenin (Ser675) and α-SMA in lung sections from bleomycin-induced fibrotic mice, treated with or without C75. n = 3. **(Q)** Representative images of immunofluorescence staining for β-catenin and α-SMA in lung sections from bleomycin-induced fibrotic mice, treated with or without C75. n = 3. **(R)** Representative images of double-labeled immunofluorescence staining for α-SMA and Lipi-Deep Red in lung sections from bleomycin-induced fibrotic mice, treated with or without C75. n = 3.
Source data are available for this figure.

polybrene overnight. 24 h post-transfection, the cells were selected in a media containing 5 ng/ml puromycin.

### Protein stability of the β-catenin assay

CHX is a bacterial toxin that disrupts protein synthesis by interfering with mRNA decoding in eukaryotic cells. By adding CHX to the cell culture medium, new protein synthesis is effectively halted, enabling the observation of the protein stability. To explore whether FASN inhibition regulated the protein stability of β-catenin, after 48 h of puromycin selection of MRC-5 cells infected with the lentivirus supernatant shFASN or NC in six-well plates, fresh complete MEM was replaced. Subsequently, at 0, 4, and 8 h, 100 μg/ml CHX was added to the medium to prevent new protein synthesis of β-catenin, and Western blotting analysis was performed to detect the protein stability rate of β-catenin. For the FASN inhibitor C75, MRC-5 cells were seeded into six-well plates ($10^5$ cells/well) and treated with 30 μM C75 for 24 h, and the fresh cell culture medium was replaced. Then, cells were treated with the same concentrations of CHX and the same duration as described above, followed by Western blotting analysis.

### Autophagic flux assay of MRC-5 cells

In addition, in a normal complete culture medium, the levels of autophagy marker proteins LC3 in cells are relatively low, which makes it challenging to observe autophagy changes. CQ inhibits autophagy flux by reducing the fusion of autophagosomes with lysosomes. Treating cells with CQ during starvation significantly increases the levels of the autophagy marker protein LC3, thereby facilitating the analysis of autophagy changes. To determine whether FASN inhibition affects autophagy in fibroblasts, MRC-5 cells ($2 \times 10^4$ cells/well) were seeded onto coverslips in 24-well plates. The GFP-RFP-LC3 plasmid was transfected into cells simultaneously with the shFASN or NC plasmid using Lipofectamine 3000, or cells were transfected with the GFP-RFP-LC3 plasmid for 24 h followed by treatment with C75 (30 μM) or not. Before the end of the experiment, cells were treated with CQ (50 μM) during starvation for 4 h, fixed with 4% paraformaldehyde, stained with DAPI working solution for 10 min, and imaged using confocal laser microscopy. For Western blotting analysis, MRC-5 cells ($10^5$ cells/well) were seeded into six-well plates, treated as described above, and then harvested for Western blotting analysis to assess the protein expression of LC3.

### Cell viability assay

The cells were seeded into 96-well plates at 100 μl containing $3 \times 10^3$ cells/well. Then, the cells were transfected with shFASN or NC for 48 h. For the CCK8 assay, 10 μl CCK8 (Beyotime) was added to each well and incubated at 37°C for 2 h, and the absorbance was measured at 450 nm by a microplate reader. In the case of the FASN inhibitor C75, MRC-5 cells were seeded into 96-well plates ($3 \times 10^3$ cells/well) and treated with 30 μM C75 for 24 h, and then, cell viability was detected by the CCK8 assay.

### Cell proliferation assay

Cell proliferation was detected by the EdU incorporation assay. Cell-Light EdU Apollo 567 In Vitro Kit was bought from Ruibo Biotechnology. After a 48 h treatment period, cells were exposed to EdU at a final concentration of 50 μM in the culture medium and incubated at 37°C for 3 h. Subsequent steps were performed according to the manufacturer's instructions, and positive cells were visualized by fluorescence microscopy.

### Transwell assay

Transwell chambers (Corning) were employed to assess cell migration and invasion. Briefly, after the designated treatment period, MRC-5 cells were suspended in serum-free MEM. Subsequently, $20 \times 10^3$ cells were seeded into the upper chamber, whereas the lower chambers were filled with MEM containing 10% FBS. After 24 h, migrated cells were stained with 0.1% crystal violet, and three random fields were imaged. Cell counts were conducted using ImageJ software.

### Wound-healing assay

The cells were cultured in six-well plates until they reached 100% confluence. Subsequently, the cell monolayers were vertically wounded using a 200 μl pipette tip, followed by gentle washing with PBS. To minimize the influence of cell growth, the culture medium was replaced with MEM containing 2% FBS. Pictures were taken at 0 h and 36 h under an inverted microscope to assess wound closure.

### Animal experiment procedure and assessment

C57BL/6N male mice (8 wk old) were procured from Vital River Laboratories. Animal care and experimental protocols were conducted in compliance with the guidelines of the Animal Care and Use Committee of Henan Normal University, adhering to the standards set forth by animal welfare organizations and national regulations. Mice were anesthetized by inhaling 40% isoflurane (diluted in 1, 2-propanediol) from RWD in Shenzhen, China, and administered 50 μl of normal saline or bleomycin (1.5 U/kg) via intratracheal injection. On day 10, the experimental group received daily intraperitoneal injections of C75 (30 mg/kg), whereas the control group received daily intraperitoneal injections of normal saline. On day 21, mice were lightly anesthetized with isoflurane for lung imaging using a Bruker SkyScan 1276 Micro-CT system (Bruker), with scanning parameters as described in a previous study (Wan et al, 2023). Subsequently, the mice were euthanized, and the total hydroxyproline content in the right lung was assessed using Hydroxyproline Assay Kit (Sigma-Aldrich) following the manufacturer's instructions. Data were reported as μg hydroxyproline/g right lung tissue.

Hematoxylin and eosin, and Masson's trichrome staining were used for lung histologic analysis (for detailed protocols, please refer to our previous studies; Wang et al, 2024). In summary, mouse lung tissues were fixed in a 4% paraformaldehyde solution for 24 h, followed by dehydration and embedding in paraffin. Four-

micrometer sections were prepared and deparaffinized to distilled water. Subsequently, hematoxylin and eosin, and Masson's tri-chrome staining were performed using staining kits from Beyotime and Solaibao Biotechnology, respectively, following the manufacturer's instructions.

## Immunofluorescence staining

Mouse or IPF paraffin-embedded tissue sections were deparaffinized in xylene followed by a gradient ethanol series, and then rehydrated in PBS. Subsequently, tissue section antigens were retrieved by immersing them in sodium citrate buffer (pH 6.0) at 95°C for 10 min, followed by gradual cooling to room temperature. For cell samples, after the respective treatments, MRC-5 cells on slides were gently washed with PBS, fixed in a 4% paraformaldehyde solution at room temperature for 10 min, neutralized with 2% glycine for 5 min, and then rinsed with PBS. Next, the sections and cell slides were permeabilized with 0.3% Triton X-100 and blocked with 5% goat serum at room temperature for 30 min. They were then incubated with specific primary antibodies (FASN [1:100], $\alpha$-SMA [1:100], $\beta$-catenin [1:100], p-$\beta$-catenin [Ser675] [1:100], PLIN2 [1:100], and FLAG [1:100]) overnight at 4°C. After washing with PBS, fluorescent secondary antibodies were added and incubated at 37°C for 1 h, followed by staining the nucleus with DAPI. Image acquisition was performed using a Zeiss microscope or a confocal laser scanning microscope (Leica).

Mouse lung tissues were cryosectioned and stained for $\alpha$-SMA as described above. After incubation with the primary antibody, the sections were washed with PBS. A 0.1 $\mu$M/liter Lipi-Deep Red probe was then applied to the lung tissue sections and incubated overnight at 4°C. The next day, after returning to room temperature, the sections were washed three times with PBS, followed by DAPI staining of the nuclei. The slides were then mounted with anti-fade mounting medium. Images were captured using confocal microscopy.

## Immunohistochemical staining

Before incubation with the secondary antibody, the procedures for immunohistochemical staining were the same as for immunofluorescence staining. Subsequently, lung sections were incubated with biotin-labeled secondary antibodies (Beyotime) at 37°C for 30 min. Then, a DAB working solution was used to develop the lung sections, followed by counterstaining the nucleus with hematoxylin. Finally, the slices were sealed using neutral gum. Stained lung sections were photographed using light microscopy.

## Quantitative RT–PCR analysis

Total RNA from MRC-5 cells, mouse lung fibroblasts, and HEK293T cells was extracted using RNeasy kits (QIAGEN) as previously described (Zhao et al, 2023). The cDNA was synthesized by a reverse transcription kit (Promega Corporation), and the concentration and the purity of cDNA were analyzed by NanoDrop. Subsequently, according to the manufacturer's instructions, quantitative RT–PCR analysis was performed using an SYBR Green kit (Yeasen) under a LightCycler 96 fluorescent quantitative PCR system (Roche). *ACTB* was used as an internal reference, and the data were calculated by a $2^{-\Delta\Delta Ct}$ method.

**Table 1. Gene-specific primer sequence for quantitative RT–PCR analysis.**

| Primer name | Oligonucleotide sequence (5'-3') |
|---|---|
| human-*FASN*-FP | 5'-TTCTACGGCTCCACGCTCTTCC-3' |
| human-*FASN*-RP | 5'-GAAGAGTCTTCGTCAGCCAGGA-3' |
| human-*PLIN2*-FP | 5'-GATGGCAGAGAACGGTGTGAAG-3' |
| human-*PLIN2*-RP | 5'-CAGGCATAGGTATTGGCAACTGC-3' |
| mouse-*FASN*-FP | 5'-CACAGTGCTCAAAGGACATGCC-3' |
| mouse-*FASN*-RP | 5'-CACCAGGTGTAGTGCCTTCCTC-3' |
| human-*β-catenin*-FP | 5'-CACAAGCAGAGTGCTGAAGGTG-3' |
| human-*β-catenin*-RP | 5'-GATTCCTGAGAGTCCAAAGACAG-3' |
| mouse-*β-catenin*-FP | 5'-GTTCGCCTTCATTATGGACTGCC-3' |
| mouse-*β-catenin*-RP | 5'-ATAGCACCCTGTTCCCGCAAAG-3' |

The forward primer (FP) and reverse primer (RP) pairs for *fibronectin*, *collagen 1*, and *ACTA2* used in this study were as described in our previous references (Wan et al, 2023). The other gene-specific primers used are listed in Table 1.

## Separation of proteins in the cytoplasm and nucleus

In short, the MRC-5 cells were digested with 0.25% trypsin and resuspended, immediately followed by lysis of the cells with nuclear and cytoplasmic separation buffer containing PMSF for 15 min on ice, then centrifuged at 1,000*g* per minute for 10 min to obtain the cytoplasm in the supernatant and nuclei in the precipitate. Next, the cytoplasm and nucleus were lysed by sodium dodecyl sulfate lysis buffer separately, and the concentration of the proteins was measured by NanoDrop.

## Immunoprecipitation

After transfecting HEK293T cells with Phage-FLAG-$\beta$-catenin, Phage-FLAG-GSK3B, Phage-FLAG-CK1a, Phage-FLAG-Axin1, and vector plasmids (15 $\mu$g) for 48 h, the cells were gently washed with ice-cold PBS. Immunoprecipitation lysis buffer containing 1:100 protein inhibitor cocktails (Beyotime) was added, and the cells were lysed on ice for 60 min. Protein concentration was measured using a BCA assay, and 20 $\mu$l of Bimake anti-FLAG magnetic beads (B26102; Bimake) was mixed with 500 $\mu$g of protein sample. The samples were then incubated on a rotator (RML-80Pro; JOANLAB) at 25 rpm/min overnight at 4°C to form immune complexes. Subsequently, the samples were collected using a magnetic holder and heated at 75°C for 15 min.

## Western blotting

Briefly, total proteins from mouse lung tissues and cultured cells were extracted using RIPA lysis buffer (Beyotime) supplemented with 1:100 PMSF. Protein concentration was determined using BCA Protein Assay Kit (Solarbio Life Sciences). The proteins were then mixed with protein loading buffer (Beyotime) and heated at 95°C for 10 min. Subsequently, 30 $\mu$g of total proteins was separated by sodium dodecyl sulfate–polyacrylamide gel electrophoresis and

**Table 2.  Primary antibodies for Western blotting.**

| Antibody name | Proportion of dilution | Item no. | Manufacturers |
|---|---|---|---|
| Anti-FASN | 1: 2,000 | 3180S | Cell Signaling Technology |
| Anti-β-catenin | 1: 2,000 | 8480S | Cell Signaling Technology |
| Anti-LC3 | 1: 2,000 | ab63817 | Abcam |
| Anti-FLAG | 1: 5,000 | 66008-4-Ig | Protein Technology |
| Anti-fibronectin | 1: 2,000 | 26836 | Cell Signaling Technology |
| Anti-collagen 1 | 1: 1,000 | 72026 | Cell Signaling Technology |
| Anti-α-smooth muscle actin | 1: 1,000 | Ab12494 | Abcam |
| Anti-β-actin | 1: 2,000 | ab8226 | Abcam |
| Anti-p-β-catenin (Ser675) | 1:2,000 | 4176 | Cell Signaling Technology |
| Anti-PLIN2 | 1:1,000 | K007677P | Solarbio Life Science |

transferred onto a polyvinylidene fluoride membrane. The membrane was blocked with 5% fat-free milk for 45 min, followed by incubation with specific primary antibodies overnight at 4°C. The concentrations of antibodies used are detailed in Table 2. After washing, the membrane was incubated with the corresponding horseradish peroxidase–conjugated secondary antibodies for about 1.5 h at room temperature. Immunoreactive bands were visualized using a chemiluminescence reagent kit purchased from Beyotime Biotechnology, and images were captured using an imager station (Gene Company Limited).

## Statistical analysis

All data were conducted for normality tests before analysis. For non-normally distributed sample data, comparisons between two groups were analyzed using the Mann–Whitney $U$ test, whereas for normally distributed data, an unpaired $t$ test was employed. Data were expressed as the mean ± SD, with statistical significance considered at $*P < 0.05$, $**P < 0.01$, $***P < 0.001$, and $****P < 0.001$.

# Data Availability

I confirm that I have included a citation for available data in my References section. The data that support the findings of this study are available from the corresponding author upon reasonable request.

## Ethics Statement

The animal handling procedures followed the protocols of the Basel Declaration and were approved by the Institutional Care and Ethics Committee of Henan Normal University (approval number: HTU 2019-02; date: 2 February 2019). Our study involving human materials conformed to the principles of the Helsinki Declaration and was approved by the Medical Research Ethics Committee of Henan Chest Hospital (approval number: 2020-03-06; date: 6 March 2020).

# Supplementary Information

# Acknowledgements

This work was supported by the Ministry of Science and Technology, PR China (2019YFE0119500); Key R&D Program of Henan Province (231111310400); Zhongyuan Scholar (244000510009); and Henan Project of Science and Technology (232102521025 and GZS2023008).

## Author Contributions

H Lian: data curation and writing—original draft, review, and editing.
Y Zhang: data curation, methodology, and writing—original draft.
Z Zhu: data curation, methodology, and writing—original draft.
R Wan: data curation, methodology, and writing—original draft.
Z Wang: methodology and writing—original draft.
K Yang: methodology and writing—original draft.
S Ma: methodology and writing—original draft.
Y Wang: methodology and writing—original draft.
K Xu: data curation, methodology, and writing—original draft, review, and editing.
L Cheng: methodology and writing—review and editing.
W Zhao: methodology and writing—review and editing.
Y Li: methodology and writing—review and editing.
L Wang: supervision, funding acquisition, project administration, and writing—review and editing.
G Yu: supervision, funding acquisition, project administration, and writing—review and editing.

## Conflict of Interest Statement

The authors declare that they have no conflict of interest.

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
