## [Reviewer comments · Life Science Alliance]

Life Science Alliance

Fatty Acid Synthase Inhibition Alleviates Lung Fibrosis via β -catenin Signal in Fibroblasts

Hui Lian, Yujie Zhang, Zhao Zhu, Ruyan Wan, Zhixia Wang, Kun Yang, Shuaichen Ma, Yaxuan Wang, Kai Xu, Lianhui Cheng, Wenyu Zhao, Yajun Li, Lan Wang, and Guoying Yu

DOI: <https://doi.org/10.26508/lsa.202402805>

Corresponding author(s): Guoying Yu, Henan Normal University and Lan Wang

Review Timeline:

Submission Date:	2024-05-03
Editorial Decision:	2024-08-05
Revision Received:	2024-11-04
Editorial Decision:	2024-11-05
Revision Received:	2024-11-11
Accepted:	2024-11-11

Transaction Report:

August 5, 2024

Re: Life Science Alliance manuscript #LSA-2024-02805

Prof. Guoying Yu
Henan Normal University
46 Jianshe Road, Xinxiang, Henan 453007, China

Dear Dr. Yu,

Thank you for submitting your manuscript entitled "Fatty acid synthase inhibition in fibroblasts alleviates lung fibrosis via regulation of β -catenin" to Life Science Alliance. The manuscript was assessed by expert reviewers, whose comments are appended to this letter. We invite you to submit a revised manuscript addressing the Reviewer comments.

Thank you for this interesting contribution to Life Science Alliance. We are looking forward to receiving your revised manuscript.

Sincerely,

B. MANUSCRIPT ORGANIZATION AND FORMATTING:

Reviewer #1 (Comments to the Authors (Required)):

In the current study Lian et al investigated mechanisms of the abnormal lipid metabolism in the pathogenesis of lung fibrosis. The topic is important and interesting however there are many confusing points should be clarified.

1. In the introduction part, studies of FASN on cancer are excess.
2. The author stated that "These studies found that compared to normal controls, both the mRNA and protein expression levels of FASN were reduced in lung tissues of IPF patients, particularly in alveolar epithelial cells, ciliated cells, and macrophages (Qian et al, 2022)." However, there is only one study being cited.
3. Details should be added to the labels of figures. Such as in Figure 1C F the label should be clearer (cell line or isolate primary cells). Otherwise, I will deem the whole lung tissue was subjected to WB analysis (Figure 1F). Please check all the figures to make them self-evident.
4. Why use cycloheximide (CHX) and chloroquine should be explained.
5. The regulation relation between FASN and β -catenin is confusing. As the author said " β -catenin could promote cell proliferation, migration, and the secretion of more cytokines and matrix proteins", If β -catenin served as an upstream regulator of FASN how could the author weight the importance of β -catenin and FASN in facilitating the activation of lung fibroblast? Why not targeting β -catenin for suppressing lung fibroblast activation?
6. The author concluded that "C75 alleviated pulmonary fibrosis in bleomycin-induced mouse lungs" by intraperitoneal injecting C75 to the mice. While one of your cited study Shin et al, 2023 demonstrated that "Overexpression of FASN attenuates bleomycin induced lung fibrosis" How can you explain this adverse function of FASN in different cells (fibroblast and alveolar epithelial cell)?
7. FASN could interact with β -catenin and GSK3B and Axin1, this is a rather complex relation. What do you consider the possible function of the interaction between FASN and β -catenin in your study?
8. In each end of the results, a conclusive sentence should be added to summarize the key points.
9. The relation among autophagy, FASN mediated β -catenin degradation and the interaction between FASN β -catenin and GSK3B in confusing. Which mechanism contribute to the FASN mediated upregulation of β -catenin?
10. In the discussion the author said "FASN is known to facilitate nucleic acid, protein, and lipid synthesis to support cancer cell metabolism" Since FASN is multifunctional in regulating nucleic acid, protein, and lipid synthesis metabolism. How could the author distinguish the possible target metabolism pathway of C75 in alleviating pulmonary fibrosis in bleomycin-induced mouse lungs.

Reviewer #2 (Comments to the Authors (Required)):

Lian et al report that FASN inhibition in MRC-5 cells or FASN pharmacological inhibition using C75 lead to inhibition of β -catenin signaling. In vivo, C75 therapeutic treatment of Bleo-treated mice led to decreased fibrosis

Figure 1: The immunofluorescence data (A and B panels) need to be quantified. Co-labeling with epithelial or mesenchymal markers should be shown.

Figure 3: bulk RNAseq upon shFASN (with the corresponding control) at different time points should be carried out to better define the general impact of FASN inhibition. Same thing with C75 vs control (bulk RNA seq).

Figure 7: what is the impact of C75 treatment on the AT2s? is it protecting the AT2s from undergoing apoptosis? Figure 7I, the authors should show staining for the activated form of β -catenin. Again co-staining with epithelial and mesenchymal markers should be performed for the IF. More markers of myofibroblasts should be examined upon the different treatments (Myh11, Acta2) as well as alveolar fibroblasts (Scube 2, Plin2, LipidTox etc..). Is C75 accelerating the activated myofibroblast to fibroblast transition during fibrosis resolution (see recent paper from Tsukui et al, Nature volume 631, pages 627-634 (2024)). What is the mechanism of action of FASN during fibrosis resolution?

Reviewer #1 (Comments to the Authors (Required)):

In the current study Lian et al investigated mechanisms of the abnormal lipid metabolism in the pathogenesis of lung fibrosis. The topic is important and interesting however there are many confusing points should be clarified.

1. In the introduction part, studies of FASN on cancer are excess.

Response: Done. Thank you for your suggestions. We have removed the part about FASN on cancer in the introduction.

2. The author stated that "These studies found that compared to normal controls, both the mRNA and protein expression levels of FASN were reduced in lung tissues of IPF patients, particularly in alveolar epithelial cells, ciliated cells, and macrophages (Qian et al, 2022)." However, there is only one study being cited.

Response: Thank you for your suggestion. Some other studies, Such as Hayek H et al, 2024 [PMID: 38117249], and Shin et al, 2023 [PMID: 37270622], have also demonstrated reduced FASN expression in IPF lung tissues. These references have now been included in the revised manuscript.

3. Details should be added to the labels of figures. Such as in Figure 1C F the label should be clearer (cell line or isolate primary cells). Otherwise, I will deem the whole lung tissue was subjected to WB analysis (Figure 1F). Please check all the figures to make them self-evident.

Response: Thank you for your suggestion. We have added detailed information regarding the cell lines and isolated primary cells used in the experiments to all figures in the revised manuscript.

4. Why use cycloheximide (CHX) and chloroquine should be explained.

Response: Cycloheximide (CHX) disrupts protein synthesis by interfering with mRNA decoding in eukaryotic cells. Adding CHX to the cell culture medium effectively halts the new synthesis of β -catenin, enabling the assessment of its protein stability. A

detailed explanation of the rationale for using CHX in this experiment can be found in the "Protein Stability of β -catenin Assay" section of the Materials and Methods. Please refer to the revised manuscript for further details. Chloroquine (CQ) inhibits autophagic flux by reducing the fusion of autophagosomes with lysosomes. Treating cells with CQ during starvation significantly increases the expression levels of the autophagy marker protein LC3, facilitating the analysis of autophagic changes. A detailed explanation of the rationale for using CQ in this experiment can be found in the "Autophagic Flux Assay of MRC-5 Cells" section of the Materials and Methods. Please refer to the revised manuscript for further details.

5. The regulation relation between FASN and β -catenin is confusing. As the author said " β -catenin could promote cell proliferation, migration, and the secretion of more cytokines and matrix proteins", If β -catenin served as an upstream regulator of FASN how could the author weight the importance of β -catenin and FASN in facilitating the activation of lung fibroblast? Why not targeting β -catenin for suppressing lung fibroblast activation?

Response: As a key enzyme in fatty acid synthesis, FASN plays a crucial role in cellular metabolism and provides a survival advantage for cells. Our experimental results indicate that β -catenin upregulates FASN expression, granting fibroblasts a survival advantage and promoting their activation. Conversely, inhibiting FASN induces a quiescent state in fibroblasts. This effect is closely linked to a decrease in β -catenin expression within the cells, as well as the loss of survival advantage following FASN inhibition. This suggests that, compared to β -catenin inhibition, FASN inhibition plays a more crucial role in regulating the activation of pulmonary fibroblasts. Therefore, we focused on inhibiting FASN in our study rather than targeting β -catenin.

6. The author concluded that "C75 alleviated pulmonary fibrosis in bleomycin-induced mouse lungs" by intraperitoneal injecting C75 to the mice. While one of your cited study Shin et al, 2023 demonstrated that "Overexpression of FASN attenuates bleomycin induced lung fibrosis" How can you explain this adverse function of FASN in different cells (fibroblast and alveolar epithelial cell)?

Response: Yes, we agree with your perspective. FASN has adverse functions in AECII cells and fibroblasts. Their opposing roles are related to their contributions during the

progression and recovery of IPF. After lung injury, AECII cells are responsible for remodeling the alveolar structure and promoting the synthesis of surfactant, thereby helping to maintain alveolar stability and gas exchange. These cells need to possess robust proliferation and differentiation capabilities to effectively replace damaged epithelial cells. In contrast, fibroblasts in injured lung tissue are often in an activated state, producing excessive extracellular matrix, which leads to lung fibrosis. Controlling fibroblast activation is a therapeutic goal for IPF. While high expression of FASN in alveolar epithelial cells can enhance mitochondrial function and alleviate lung fibrosis, it also promotes the survival and activation of fibroblasts, contributing to the progression of fibrosis. Knocking out FASN in alveolar epithelial cells impairs AECII stemness and exacerbates fibrosis, whereas knocking out FASN in fibroblasts induces them into a quiescent state, thereby alleviating lung fibrosis. Ensuring that FASN expression is mutually coordinated between these two cell types could help improve the pathological state of IPF.

7. FASN could interact with β -catenin and GSK3B and Axin1, this is a rather complex relation. What do you consider the possible function of the interaction between FASN and β -catenin in your study?

Response: As a key enzyme involved in de novo fatty acid synthesis, increased expression of FASN provides metabolic resources for rapid cell growth. Therefore, the elevation of FASN expression driven by β -catenin meets the metabolic demands necessary for sustaining β -catenin activity and the associated fibrotic processes, leading to continuous activation of fibroblast growth and invasiveness, thereby accelerating the progression of lung fibrosis. Additionally, increased FASN expression supplies the lipids required for membrane and cytoplasmic remodeling, enhancing fibroblast motility while also increasing the stability of β -catenin in the cytoplasm and supporting its nuclear translocation. Consequently, the interaction between FASN and β -catenin in fibroblasts creates a favorable environment for fibroblast activation.

8. In each end of the results, a conclusive sentence should be added to summarize the key points.

Response: Done, thank you for your suggestion.

transition (EMT) in distal lung epithelial stem cells (i.e., AT2 cells), all of which can contribute to the dysregulation and pathogenic activation of fibroblasts. We treated AT2 cells (A549 cells) with C75, resulting in significant reductions in the protein levels of EMT-related markers, including N-Cadherin, E-Cadherin, and Vimentin, as well as the senescence-associated markers P21 and P53. These findings suggest that C75 treatment can effectively inhibit EMT and senescence in AT2 cells. Please refer to the figure below.

However, previous experimental studies have shown that the expression of FASN in AT2 cells within IPF lung tissue is significantly reduced. This decrease in FASN expression can lead to a loss of mitochondrial membrane potential and the generation of reactive oxygen species (ROS) in AT2 cells. Therefore, we speculate that C75 treatment may also induce mitochondrial dysfunction in these cells. Nevertheless, since FASN expression is already diminished in AT2 cells within IPF lung tissue, C75 treatment may not further exacerbate mitochondrial dysfunction to the extent that it induces apoptosis. Consequently, we believe that the effects of C75 treatment on AT2 cells are complex, and its specific impacts warrant further investigation.

4. Figure 7I, the authors should show staining for the activated form of β -catenin. Again co-staining with epithelial and mesenchymal markers should be performed for the IF. More markers of myofibroblasts should be examined upon the different treatments (Myh11, Acta2) as well as alveolar fibroblasts (Scube 2, Plin2, LipidTox etc..).

Response: Phospho- β -catenin (Ser675) is an activated form of β -catenin that accumulates in the nucleus, enhancing transcriptional activity. We used immunofluorescent co-staining for Phospho- β -catenin (Ser675) and α -SMA to assess its expression. (Refer to Figure 7P).

As the antibody for Phospho- β -catenin (Ser675) is rabbit-derived, and after thorough investigation, we found that the epithelial cell marker SP-C's primary antibody is also derived from rabbit. Unfortunately, we could not obtain a primary antibody for SP-C from another species. Using two primary antibodies from the same species for

9. The relation among autophagy, FASN mediated β -catenin degradation and the interaction between FASN β -catenin and GSK3B in confusing. Which mechanism contribute to the FASN mediated upregulation of β -catenin?

Response: Our experimental results indicate that inhibiting FASN reduces β -catenin expression in fibroblasts. Based on these findings, β -catenin can be degraded via the autophagy-lysosomal pathway following FASN inhibition, and it may also be decreased through a direct interaction between FASN and β -catenin. However, we did not conduct overexpression experiments for FASN, and we speculate that FASN may promote β -catenin expression through direct interaction, as well as upregulate its levels by inhibiting autophagic degradation. Further experimental validation is needed to confirm this hypothesis.

10. In the discussion the author said "FASN is known to facilitate nucleic acid, protein, and lipid synthesis to support cancer cell metabolism" Since FASN is multifunctional in regulating nucleic acid, protein, and lipid synthesis metabolism. How could the author distinguish the possible target metabolism pathway of C75 in alleviating pulmonary fibrosis in bleomycin-induced mouse lungs.

Response: Previous studies have identified lipid metabolism dysregulation as one of the major metabolic changes in IPF. Dysregulation of lipid metabolism in various cell types, including alveolar type II (ATII) cells, alveolar macrophages, and fibroblasts, drives the pathogenesis of pulmonary fibrosis. As an inhibitor of FASN, C75 primarily acts to inhibit FASN enzyme activity. Therefore, during the treatment of bleomycin-induced pulmonary fibrosis in mice, the lipid metabolism pathway emerges as an important target pathway.

The PI3K/AKT/mTOR signaling pathway is involved in regulating cell proliferation, differentiation, apoptosis, and autophagy. Among these, mTOR is a central molecule that regulates growth and metabolism, responding to insulin signaling to modulate glucose metabolism, as well as regulating amino acid metabolism to balance cellular nutrient metabolism. To investigate whether C75 is involved in regulating the metabolism of mice with pulmonary fibrosis, we treated MRC-5 cells with C75 in vitro. The results indicated that C75 treatment significantly reduced the protein expression of p-AKT and p-mTOR compared to the control group, suggesting that C75 has the

potential to alter the metabolic pathways in mice with pulmonary fibrosis (Refer to the picture below).

Research has shown that several amino acids, including glutamine, glycine, arginine, and proline, play roles in pulmonary fibrosis. Glycine is one of the primary amino acids involved in collagen formation. During TGF- β -stimulated fibroblasts, the expression of various enzymes involved in the de novo biosynthetic pathway of glycine is upregulated. This upregulation can be attributed to the mTOR-ATF4 axis. Since C75 treatment reduced p-mTOR protein expression in fibroblasts, we hypothesize that C75 may be involved in regulating glycine biosynthesis.

Several studies have defined glutamine as an important promoter of pulmonary fibrosis, with its mechanism enhancing myofibroblast differentiation and collagen synthesis. Research on pulmonary myofibroblasts indicates that increased glutamine catabolism can enhance collagen production by activating mTOR complex 1. Given that C75 treatment reduces p-mTOR protein expression in fibroblasts, we speculate that following glutamine catabolism, the intermediate signaling molecules that promote collagen synthesis may be lost, thus exerting an inhibitory effect on pulmonary fibrosis.

Furthermore, some studies have linked glycolytic dysregulation to IPF, showing that the upregulation of glycolytic enzymes, particularly glucose transporter proteins (GLUTs), especially GLUT1, and the subsequent increase in glucose uptake are mediated by TGF- β . Extensive research has been conducted on the mechanisms underlying GLUT1 upregulation, including SMAD2/3-dependent and -independent pathways, as well as the PI3K/AKT/mTOR pathway. Since C75 treatment reduced p-AKT and p-mTOR protein expression in fibroblasts, we hypothesize that following C75 treatment, GLUT1 expression may be reduced, leading to decreased glucose uptake and involvement in glucose metabolism.

Reviewer #2 (Comments to the Authors (Required)):

Lian et al report that FASN inhibition in MRC-5 cells or FASN pharmacological inhibition using C75 lead to inhibition of β -catenin signaling. In vivo, C75 therapeutic treatment of Bleo-treated mice led to decreased fibrosis

1. Figure 1: The immunofluorescence data (A and B panels) need to be quantified. Co-labeling with epithelial or mesenchymal markers should be shown.

Response: The number of FASN and α -SMA co-labeled cells in Figures 1A and 1B has been quantified. Please refer to the revised Figure 1.

2. Figure 3: bulk RNAseq upon shFASN (with the corresponding control) at different time points should be carried out to better define the general impact of FASN inhibition. Same thing with C75 vs control (bulk RNA seq).

Response: Thank you for your suggestion regarding bulk RNA sequencing (RNAseq) to assess the broader impact of FASN inhibition. While we appreciate the insights RNAseq could provide, our study's primary focus is on the mechanistic understanding of FASN inhibition. We have already conducted detailed functional assays, including immunofluorescence for Collagen1 and Perilipin (PLIN2), Western blot analysis of Fibronectin, Collagen1, α -SMA, and PLIN2, as well as RT-qPCR for FN1, COL1A1, ACTA2 and PLIN2. These experiments provide substantial evidence of the effects of shFASN and C75 treatment on key fibrotic pathways, confirming FASN's role in regulating these processes. Given the comprehensive data from these targeted assays, we believe the current conclusions are well-supported without the need for RNAseq. However, we acknowledge that RNAseq could be a valuable follow-up study for more in-depth exploration of the broader transcriptomic changes, which could be pursued in future research.

3. Figure 7: what is the impact of C75 treatment on the AT2s? is it protecting the AT2s from undergoing apoptosis?

Response: In the pathogenesis of IPF, chronic injury leads to mitochondrial dysfunction, endoplasmic reticulum stress, cellular senescence, and epithelial-mesenchymal

immunofluorescent double staining could result in false positives; therefore, we did not perform double staining for SP-C and Phospho- β -catenin (Ser675).

Additionally, under various treatment conditions, we further examined the protein levels of the myofibroblast marker α -SMA and the ECM-related protein Collagen1 (Refer to Figure 7H-K), as well as the expression of the adipofibroblast markers Plin2 (Refer to Figure 7H, L and O).

The Lipi-Deep Red probe was used to detect lipid droplet content in mouse lung tissues. Compared to the saline group, the fibrotic regions in the bleomycin group showed a significant reduction in lipid droplet content, while lipid droplet levels were relatively increased in bleomycin-treated lung tissues after C75 treatment (Figure 7R).

5. Is C75 accelerating the activated myofibroblast to fibroblast transition during fibrosis resolution (see recent paper from Tsukui et al, Nature volume 631, pages 627-634 (2024)).

Response: We reviewed the work by Tsukui et al., which highlights the role of alveolar fibroblasts in supporting alveolar stem cells and moderating responses to acute lung injury. They showed that blocking fibroblast induction in the alveolar fibroblast lineage could suppress fibrosis, with alveolar fibroblasts marked by Plin2, indicative of a quiescent adipofibroblast phenotype. During fibrosis resolution, their study found that activated myofibroblasts dedifferentiate back into alveolar fibroblasts. Building on this, our additional in vitro and in vivo experiments demonstrate that C75 treatment not only downregulates fibronectin, collagen 1, and α -SMA but also increases expression of the alveolar fibroblast marker Plin2 (Figure 7H, O). This suggests that C75 may indeed promote the transition of activated myofibroblasts to alveolar fibroblasts, contributing to lung fibrosis resolution. We have incorporated this interpretation into the Discussion to further clarify C75's role in facilitating fibrosis resolution.

6. What is the mechanism of action of FASN during fibrosis resolution?

Response: Given that the fibrotic response triggered by bleomycin treatment is reversible, researchers have used this model to study the fate of activated myofibroblasts after fibrosis resolution. The results showed that during fibrosis resolution, activated myofibroblasts and their progeny are not eliminated from the lungs, with some cells converting to an adipofibroblast-like phenotype. PPAR γ serves as a master regulator for the lipogenic differentiation of preadipocytes and mesenchymal

stem cells and is also involved in the formation of adipofibroblasts. PPAR γ agonists have been shown to protect mice from developing fibrosis.

Researchers speculate that β -catenin-mediated fibrosis may target the progenitor cell types that give rise to fibroblasts, rather than targeting mature fibroblasts. We now consider this immature fibroblast type to be alveolar fibroblasts. Studies have indicated that β -catenin signaling inhibits adipogenesis by suppressing the expression of the adipogenic transcription factors CCAAT/enhancer-binding protein α (C/EBP- α) and PPAR γ . The reduction of the adipose layer observed in the fibrotic skin of systemic sclerosis patients suggests that fibrosis may occur at the expense of fat loss. Supporting this notion, mouse models have demonstrated that mice expressing Wnt10b driven by fatty acid-binding protein 4 (FABP4) exhibit a gradual decrease in subcutaneous and visceral fat tissue, accompanied by dermal fibrosis characterized by increased collagen deposition, fibroblast activation, and myofibroblast accumulation. Moreover, fibroblasts extracted from these mice exhibited enhanced Wnt signaling, along with elevated levels of collagen1 and α -SMA expression. Other studies have found that Wnt/ β -catenin signaling can inhibit adipogenic gene expression and promote dedifferentiation into a myofibroblast phenotype in liver adipofibroblasts and 3T3-L1 cells. In human colorectal cancer cell lines LS174T and Caco2, activation of the Wnt/ β -catenin signaling pathway decreased the expression of PPAR γ and its target genes FABP2 and PLIN2; conversely, inhibition of Wnt signaling led to increased expression of PPAR γ and its target genes FABP2 and PLIN2. Through in vitro and in vivo experiments, we found that inhibiting FASN in fibroblasts reduced β -catenin expression. Therefore, we speculate that the reduction of β -catenin may activate PPAR γ , promoting the dedifferentiation of myofibroblasts into adipofibroblasts. We have included these details in the Discussion to elucidate the mechanism by which FASN contributes to fibrosis resolution.

November 5, 2024

RE: Life Science Alliance Manuscript #LSA-2024-02805R

Prof. Guoying Yu
Henan Normal University
46 Jianshe Road, Xinxiang, Henan 453007, China

Dear Dr. Yu,

Thank you for submitting your revised manuscript entitled "Fatty Acid Synthase Inhibition Alleviates Lung Fibrosis via β -catenin Signal in Fibroblasts". We would be happy to publish your paper in Life Science Alliance pending final revisions necessary to meet our formatting guidelines.

- please be sure that the authorship listing and order is correct
- please add ORCID ID for second corresponding author-they should have received instructions on how to do so
- please use the [10 author names, et al.] format in your references (i.e. limit the author names to the first 10)
- please upload your table files as editable doc or excel files
- please add a figure callout for Figure 7A to the main manuscript text
- please upload your main figures as single files; these will be displayed in-line in the HTML version of your paper, so please provide them as single page files (Figure 7 currently spans 2 pages); we do not have a limit on the number of main figures and these can be split if necessary for space

Figure Check:

- please add sizes next to all blots

A. FINAL FILES:

B. MANUSCRIPT ORGANIZATION AND FORMATTING:

Sincerely,

November 11, 2024

RE: Life Science Alliance Manuscript #LSA-2024-02805RR

Prof. Guoying Yu
Henan Normal University
46 Jianshe Road, Xinxiang, Henan 453007, China

Dear Dr. Yu,

Thank you for submitting your Research Article entitled "Fatty Acid Synthase Inhibition Alleviates Lung Fibrosis via β -catenin Signal in Fibroblasts". It is a pleasure to let you know that your manuscript is now accepted for publication in Life Science Alliance. Congratulations on this interesting work.

DISTRIBUTION OF MATERIALS:

Again, congratulations on a very nice paper. I hope you found the review process to be constructive and are pleased with how the manuscript was handled editorially. We look forward to future exciting submissions from your lab.

Sincerely,
